# video-SALMONN S: Memory-Enhanced Streaming Audio-Visual LLM

**Guangzhi Sun** [1 2]   **Yixuan Li** [1]   **Xiaodong Wu** [2]   **Yudong Yang** [1]   **Wei Li** [3]   **Zejun Ma** [3]   **Chao Zhang** [1]

## Abstract

Long-duration streaming video understanding is fundamental for future AI agents, yet remains limited by ineffective long-term memory. We introduce video-SALMONN S, a memory-enhanced streaming audio–visual large language model that processes over 3-hour videos at 1 FPS and 360p resolution, outperforming strong non-streaming models under the same memory budget. In addition to token merging or downsampling, video-SALMONN S is the first to employ *test-time training* (TTT) as a streaming memory mechanism for video understanding. TTT continuously transforms short-term multimodal representations into long-term memory embedded in model parameters. To improve long-range dependency modeling and memory capacity, we propose (i) a $\text{TTT}_{\text{MEM}}$ layer with an additional long-span prediction objective, (ii) a two-stage training scheme, and (iii) a modality-aware memory reader. We further introduce the **e**pisodic **l**earning from **vi**deo **m**emory (ELViM) benchmark, simulating agent-like scenarios where models must learn from videos observed hours earlier. video-SALMONN S consistently outperforms both streaming and non-streaming baselines by 3-7% on long video benchmarks. Notably, video-SALMONN S achieves a $15\%$ absolute accuracy improvement over strong non-streaming models on ELViM, demonstrating strong learning abilities from video memory.[1]

## 1. Introduction

Processing video streams of any length at a fixed high frame rate and a decent resolution is one of the crucial abilities

in future AI agents. Despite the rapid progress in state-of-the-art (SOTA) video large language models (LLMs) (Li et al., 2024; Zhang et al., 2024c; Wang et al., 2024a; Lin et al., 2024b; Bai et al., 2025b; Zhang et al., 2025a; Tang et al., 2025a; Sun et al., 2025), most of them only perform offline video understanding by setting a maximum number of input visual tokens to the Transformer, hence yielding a significant information loss for long videos. Some recent work explores token compression methods to achieve longer video understanding (Li et al., 2025; Tan et al., 2024; Tang et al., 2025b; Chen et al., 2024; Zhang et al., 2025c), especially ones using the user prompt to achieve high compression ratios (Gao et al., 2024; Wang et al., 2025b).

In real-world scenarios, where both video length and the timing of user queries are typically unknown, offline formulations become challenging to deploy due to the limited attention span of Transformer architectures. This has led to streaming video understanding models that operate at a fixed frame rate and maintain bounded memory via token merging or discarding (Qian et al., 2025; Yang et al., 2025b; Song et al., 2024; Zhang et al., 2025b; Huang et al., 2025; Zeng et al., 2025). While effective for short streams, these methods incur cumulative information loss over long durations, often underperforming relative to offline models. Recent extensions leverage agentic frameworks with external memory (Long et al., 2025) or task-specific frame selection, such as for spatial reasoning (Yang et al., 2025a), but typically require specialized training data or complex memory designs, limiting scalability and practical deployment.

To address these limitations, we propose video-SALMONN S, a streaming audio-visual LLM that achieves SOTA performance on videos exceeding 3 hours in length at 1 FPS and 360p resolution under a fixed memory budget. To the best of our knowledge, video-SALMONN S is the first streaming video LLM to incorporate test-time training (TTT) as a mechanism for enhancing memory. Unlike existing memory consolidation approaches, TTT enables online adaptation through fast-weight updates, allowing model parameters to function as an implicit, parameter-based memory for integrating long-term information. Specifically, we propose the $\text{TTT}_{\text{MEM}}$ layer, which extends traditional TTT formulations (Sun et al., 2024; Dalal et al., 2025) by incorporating a long-span prediction objective to strengthen long-context modeling. The $\text{TTT}_{\text{MEM}}$ layer is trained using a two-stage

---

[1]Tsinghua University [2]University of Cambridge [3]ByteDance. Correspondence to: Chao Zhang <cz277@tsinghau.edu>.

*Proceedings of the 43rd International Conference on Machine Learning*, Seoul, South Korea. PMLR 306, 2026. Copyright 2026 by the author(s).

[1]https://github.com/bytedance/SALMONN/tree/video-salmonn-S

strategy: a cold-start stage that optimizes static TTT parameters with medium-length contexts, followed by a scale-up stage that progressively expands both context length and memory capacity. At last, we employ a modality-aware, prompt-dependent memory reading mechanism that recurrently compresses multimodal KV caches while strictly adhering to streaming constraints with constant memory usage.

To reflect the practical requirement of long-term memory in video-based agents that operate continuously over streaming inputs, we introduce the **e**pisodic **l**earning from **vi**deo **m**emory (ELViM) benchmark. ELViM is the first benchmark explicitly designed to evaluate whether streaming video LLMs can acquire procedural knowledge from past video experiences and reuse it for real-time decision-making. In contrast to existing video benchmarks that focus primarily on short-term perception or within-clip reasoning, ELViM directly assesses the ability to consolidate, retain, and recall information from videos observed hours earlier under realistic streaming constraints. Main contributions are summarised as follows:

- We propose video-SALMONN S, a SOTA streaming audio-visual LLM and the first to leverage TTT as a streaming memory mechanism, enabling over 3 hours of video understanding at 1 FPS and 360p under a fixed memory budget through a novel $TTT_{MEM}$ layer, tailored training loss and scheme and memory-reading designs.

- We propose ELViM, the first benchmark designed to explicitly evaluate whether streaming video LLMs can learn procedural knowledge from past video memory. ELViM provides a principled evaluation for long-term memory and continual learning in streaming video LLMs.

- video-SALMONN S is evaluated on a range of online and general video understanding benchmarks. In particular, video-SALMONN-S consistently exceeds strong streaming and non-streaming baselines by 3-7% in absolute accuracy on long video benchmarks, with 15% accuracy improvement on ELViM.

## 2. Related Work

### 2.1. Long Video Understanding

The key challenge of long video understanding is efficient compression of information into a size that the LLM can process. A number of training-based methods (Li et al., 2025; Zhang et al., 2024b; Shen et al., 2024; Shu et al., 2024; Liu et al., 2025a) have been proposed to reduce the number of visual tokens needed to represent each video frame. Some recent visual LLMs, such as Qwen2.5-VL (Bai et al., 2025b), inherently contain token merging modules and a much longer context LLM backbone that are suitable for long video understanding. Meanwhile, training-free methods (Liu et al., 2025b; Zhang et al., 2024d; Yang

et al., 2025b; Wang et al., 2024c; 2025b) are also investigated, which mainly select the key-value (KV) cache in each Transformer block. Specifically, ReTaKe (Wang et al., 2024c) applies a dynamic KV-cache compression by selecting and keeping the important KV-pairs only based on their similarities. The follow-up work, AdaReTaKe (Wang et al., 2025b) applies prompt-based KV-Cache selection depending on the attention scores.

### 2.2. Online Video Understanding and TTT

Online video understanding requires processing frames continuously in real time, where the stream length is unknown, precluding the offline practice of pre-selecting a fixed, uniformly sampled frame set. Models, therefore, must operate under a fixed memory budget to stay within the LLM's context window. Prior work follows two main directions. Token-reduction methods fix the number of visual tokens: MovieChat (Song et al., 2024) merges tokens via similarity-based consolidation, and VideoLLMonline (Chen et al., 2024) reduces each frame to about 10 tokens for efficiency. External-memory methods compress and organise visual tokens and retrieve the most relevant ones at query time: Flash-VStream (Zhang et al., 2024a) and Dispider (Qian et al., 2025) fuse retrieved visual tokens with text tokens before feeding them into the LLM backbone. A complementary line targets KV-cache compression and retrieval: ReKV (Di et al., 2025) offloads layer-wise caches to external storage for on-demand fetching; (Ning et al., 2025) compresses the KV cache to cut memory and accelerate question answering (QA) relative to ReKV; StreamMem and InfiniPot-V (Kim et al., 2025) further improve efficiency via dynamic, non-uniform KV compression. Wang et al. (2025a) explores test-time training for video segmentation, but focuses on adaptation to local changes at test time that do not require long-term dependencies. In contrast, **video-SALMONN S** mitigates hard token dropping/merging by using TTT to continually update the memory representations, mitigating information loss while meeting a fixed memory budget.

## 3. video-SALMONN S Training and Test

The overall structure of video-SALMONN S is shown in Fig. 1. The input video is processed in streaming mode at a fixed frame rate, *e.g.* 1 FPS. Each video frame is first converted into video encodings using a visual encoder, followed by a pre-trained modality aligner that projects the representations to the text space, denoted as $\mathbf{X}_t$. Video encodings $\mathbf{X}_t$ are then fed into the $TTT_{MEM}$ layer, which incorporates the full sequence history into the representations. We maintain a long-term memory of a fixed number of tokens, and the output of the $TTT_{MEM}$ layer for the current frame provides new incoming tokens to be added to the memory. Thereafter, merging or downsampling is performed to ensure a fixed

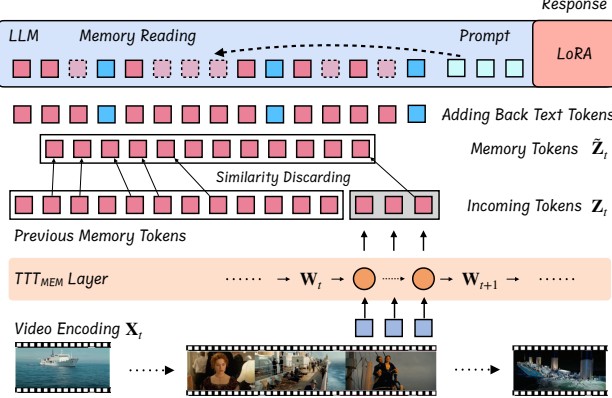

*Figure 1.* Overall model structure of video-SALMONN S. The video encodings are first passed through the $\text{TTT}_{\text{MEM}}$ layer (Section 3.1), followed by a similarity discarding procedure to keep fixed-size memory. The fixed memory is then used as the input to the LLM, optionally using a modality-aware prompt-dependent reading mechanism (Section 3.3).

number of tokens is sent to the Transformer.

In recent LLM designs, such as Qwen3-VL (Bai et al., 2025a), text and timestamp tokens are often interleaved with multimodal tokens. We use $\text{TTT}_{\text{MEM}}$ to process multimodal tokens only, and then add those text tokens back while maintaining their original relative positions in the sequence. Besides, to enable a larger memory size while still maintain fixed GPU footprint, video-SALMONN S employs a modality-aware prompt-dependent memory reading mechanism in each Transformer block to select useful KV-cache based on attention scores. The $\text{TTT}_{\text{MEM}}$ layer and the low-rank adapter (LoRA) contain trainable parameters, and other parts of the model are frozen throughout training.

### 3.1. Memory with $\text{TTT}_{\text{MEM}}$ Layer

The long-term memory in video-SALMONN contains a $\text{TTT}_{\text{MEM}}$ layer followed by a token discarding process to keep the total number of tokens to the Transformer constant. Specifically, given the chunk of video encodings denoted as $\mathbf{X}_t \in \mathbb{R}^{K \times d}$ where $K$ is the number of tokens, the incoming memory token to be added to the memory is derived as

$$\mathbf{Z}_t, \mathbf{W}_t = \text{TTT}_{\text{MEM}}(\mathbf{X}_t, \mathbf{X}_{t-T}, \mathbf{W}_{t-1}), \qquad (1)$$

where $\mathbf{W}_{t-1}$ is the weight carrying history information and is updated to incorporate the information in $\mathbf{X}_t$ as well as $\mathbf{X}_{t-T}$ via a long-span prediction task. The gating mechanism following (Dalal et al., 2025) is used. Since audio tokens are only about 1/75 the number of visual tokens, they bypass $\text{TTT}_{\text{MEM}}$ and are fed directly into the token discarding stage, simplifying $\text{TTT}_{\text{MEM}}$ to visual-only inputs.

The memory tokens $\mathbf{Z}_t$ are then combined with previous memory tokens $\tilde{\mathbf{Z}}_{t-1} \in \mathbb{R}^{N \times d}$ where $N$ is the fixed number of memory tokens following a cosine similarity-based

token discarding procedure (Song et al., 2024; Yang et al., 2025b). Specifically, let us denote $\mathbf{Z}'_t = \text{Concat}(\tilde{\mathbf{Z}}_{t-1}, \mathbf{Z}_t)$ along the sequence dimension, that is, $\mathbf{Z}'_t \in \mathbb{R}^{(N+K) \times d}$. We discard $K$ tokens that have the highest cosine similarities with their next tokens, *i.e.* $\cos(Z'_{t,n}, Z'_{t,n+1})$. The remaining $N$ tokens become the new memory tokens, $\tilde{\mathbf{Z}}_t$. Note that we particularly chose to discard rather than the merging operation adopted in MovieChat (Song et al., 2024) to better exploit the long-term context representation ability of $\text{TTT}_{\text{MEM}}$ layer and avoid over-smoothing in extremely long videos. Despite discarding tokens, the fast-weight state produced by $\text{TTT}_{\text{MEM}}$ can preserve useful summaries of past inputs. The long-span objective encourages retention of information relevant across distant timestamps.

$$
\begin{array}{ccccc}
\mathbf{Z}_1 & & \mathbf{Z}_{t-1} & \mathbf{Z}_t = f(\mathbf{X}_t; \mathbf{W}_t) & \text{Output tokens} \\
\uparrow & & \uparrow & \uparrow & \\
\mathbf{W}_0 \longrightarrow \mathbf{W}_1 \longrightarrow & \ldots & \longrightarrow \mathbf{W}_{t-1} \longrightarrow & \mathbf{W}_t = \mathbf{W}_{t-1} - \eta \nabla l(\mathbf{W}_{t-1}; \mathbf{X}_t, \mathbf{X}_{t-T}) \\
\uparrow & & \uparrow & \uparrow & \\
\mathbf{X}_1 & & \mathbf{X}_{t-1} & \mathbf{X}_t & \text{Input tokens}
\end{array}
$$

*Figure 2.* The workflow of the $\text{TTT}_{\text{MEM}}$ layer. The layer works as a recurrent neural network, which updates the current fast-weight $\mathbf{W}_{t-1}$ of an MLP model for an incoming mini-batch of tokens $\mathbf{X}_t$ to minimise the reconstruction loss and the long-span prediction loss, which involves $\mathbf{X}_{t-T}$ as shown in Eqn. (4).

The overall workflow is shown in Fig. 2. The $\text{TTT}_{\text{MEM}}$ layer consists of a multi-layer perceptron (MLP) model with fast weights $\mathbf{W}$. We divided the entire input sequence into fixed-size chunks $\mathbf{X}_1, \mathbf{X}_2, \ldots \mathbf{X}_t, \ldots$, treating them as minibatches and sending each of them in consecutive order to the $\text{TTT}_{\text{MEM}}$ layer. Each minibatch contains around 12-16 frames. $\text{TTT}_{\text{MEM}}$ adopts the multiview reconstruction loss and the long-span prediction loss:

$$l_{\text{recon}}(t) = \|f(\theta_{\mathbf{K}}\mathbf{X}_t; \mathbf{W}_{t-1}) - \theta_{\mathbf{V}}\mathbf{X}_t\|_2 \qquad (2)$$

$$l_{\text{long-span}}(t) = \|f(\theta'_{\mathbf{K}}\mathbf{X}_t; \mathbf{W}_{t-1}) - \theta'_{\mathbf{V}}\mathbf{X}_{t-T}\|_2 \qquad (3)$$

where $\theta$ are trainable projection weights which follow the standard TTT-MLP (Sun et al., 2024) implementation to establish the multi-view reconstruction loss, and $f(\cdot; \mathbf{W})$ denotes the function of the MLP model following the residual and layer norm (LN) operation by (Dalal et al., 2025). Specifically, $\theta_{\text{QKV}} \in \mathbb{R}^{D_K \times (HD_W)}$ are trainable projection matrices that project inputs $\mathbf{X}$ into the space of $\mathbf{W}$, where $D_K$ is the dimension of key vectors, $D_W$ is the dimension of fast weights, and $H$ is the number of heads in TTT (i.e. different number of fast weight matrices). The fast weights are trained to minimize the following:

$$l(\mathbf{X}_t, \mathbf{X}_{t-T}; \mathbf{W}_{t-1}) = l_{\text{recon}}(t) + l_{\text{long-span}}(t), \qquad (4)$$

where $T$ is a hyper-parameter is the number of minibatches before the current step $t$ that we predict for the long-span loss. While $\mathbf{W}_t$ already incorporates information of past

inputs, when the sequence becomes longer, previous information would inevitably be overwritten by new updates. The long-span prediction loss acts as a temporal consistency regularizer on the fast-weight updates, encouraging the model to maintain representations predictive of inputs from $t - T$, which improves retention over long sequences.

The fast weight can be updated with gradient descent $\mathbf{W}_t = \mathbf{W}_{t-1} - \eta \nabla l(\mathbf{X}_t, \mathbf{X}_{t-T}; \mathbf{W}_{t-1})$. The updated multilayer perceptron (MLP) is used to produce the output token $\mathbf{Z}_t$ by

$$\mathbf{Z}_t = f(\theta_\mathbf{Q} \mathbf{X}_t; \mathbf{W}_t). \tag{5}$$

The name of TTT comes from the fact that the fast weight $\mathbf{W}_t$ is updated for each incoming mini-batch at test time.

### 3.2. Two-Stage Training Scheme

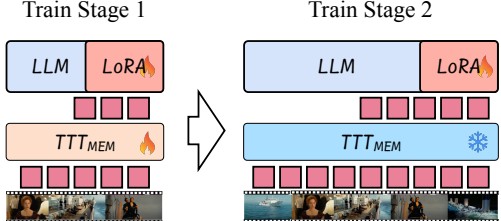

*Figure 3.* Illustration of the two-stage training scheme for video-SALMONN S. Stage 1 trains the entire model end-to-end with both LoRA and TTT$_{\text{MEM}}$ parameters updated. Stage 2 scales up with longer sequences and more memory tokens by freezing TTT$_{\text{MEM}}$ parameters while keeping the same update rule for fast weights.

One bottleneck during training is the GPU memory consumption at the TTT$_{\text{MEM}}$ layer when the input sequence gets longer. In order to balance the training of the TTT$_{\text{MEM}}$ model parameters, *i.e.* $\theta$ projection matrices in Eqn. (3), input lengths and memory tokens, the two-stage training scheme is adopted, as shown in Fig. 3. *Stage 1* is a cold-start stage where TTT$_{\text{MEM}}$ parameters are randomly initialized and are jointly trained with the LLM. After stage 1, the fast weight update mechanism is established. *Stage 2* training is conducted thereafter to scale up input and memory by freezing $\theta$ in TTT$_{\text{MEM}}$ but maintaining the fast weight update rule. This stage further optimizes the LLM to extract information from memory with much longer input sequence lengths and more memory tokens, and is critical when a higher memory budget is allowed during inference.

### 3.3. Modality-Aware Memory Reading

The video memory token sequence $\tilde{\mathbf{Z}} \in \mathcal{R}^{N \times d}$ after TTT$_{\text{MEM}}$ already contains comprehensive information from the preceding video. However, when responding to a specific prompt $\mathbf{P} \in \mathcal{R}^{S \times d}$, it is usually unnecessary to utilise all memory tokens, which brings both performance issues and high computational cost. Therefore, we employ a

modality-aware prompt-dependent memory reading mechanism to select only the relevant part of the memory.

We aim to leverage the LLM to compress the number of KV pairs used for final decoding to an average of $M$ tokens per layer. Specifically, the tokens are first divided into chunks of length $m$, resulting in $\lceil N/m \rceil$ chunks, $\tilde{\mathbf{Z}}_i$, to reduce computational requirements. For each chunk at layer $l$, we concatenate the input of this chunk with prompt tokens forwarded to layer $l$, and apply multi-head self-attention by

$$\mathbf{O}_i^{(l)} = \text{Attention}\left([\mathbf{X}_i^{(l)}; \mathbf{X}_\mathrm{p}^{(l)}] \big| \mathbf{KV}_{1:i-1}^{(l)}\right), \tag{6}$$

where $[\cdot; \cdot]$ denotes concatenation of vectors along sequence direction, $\mathbf{X}_i^{(l)}$ are $\tilde{\mathbf{Z}}_i$ forwarded to layer $l$ and $\mathbf{X}_\mathrm{p}^{(l)}$ are $\mathbf{P}$ forwarded to layer $l$. $\mathbf{KV}_{1:i-1}$ are compressed KV cache carried over up to chunk $i$. Attention $(\cdot|\cdot)$ is the standard LLM attention (Bai et al., 2025a) with KV-Cache, with $[\mathbf{X}_i^{(l)}; \mathbf{X}_\mathrm{p}^{(l)}]$ being the input. We then compute the average attention score from the prompt to each position in chunk $i$ to reflect the importance of each KV pair as

$$\mathbf{c}_i^{(l)} = \sum_{s=1}^{S} \frac{1}{H} \sum_{h=1}^{H} \mathbf{A}_i^{(l)}[h, s], \tag{7}$$

where $\mathbf{A}_i^{(l)}[h, s] \in \mathbb{R}^m$ is the attention score from token $s$ to all vectors in $\mathbf{X}_i^{(l)}$ and $H$ is the number of attention heads. We concatenate $\mathbf{c}_i^{(l)}$ with previous scores $\mathbf{c}_{1:i-1}^{(l)}$, and select a maximum of $m$ KV pairs with the highest importance for tokens that correspond to multimodal input positions, with a mask operation as follows:

$$\mathbf{I}_i = \text{ArgTopK}(\text{Mask}([\mathbf{c}_{1:i-1}^{(l)}; \mathbf{c}_i^{(l)}]), k = m) \tag{8}$$

$$\hat{\mathbf{I}}_i = \text{Sorted}([\mathbf{I}_i; \mathbf{I}_{\text{other}}]) \tag{9}$$

$$\mathbf{KV}_{1:i}^{(l)} = [\mathbf{KV}_{1:i-1}^{(l)}; \mathbf{KV}_i^{(l)}][\hat{\mathbf{I}}_i], \tag{10}$$

where $\text{Mask}(\cdot)$ is the function that selects positions corresponding to multimodal tokens, $\text{ArgTopK}(\cdot, k = \cdot)$ selects the top $k$ elements and $\text{Sorted}(\cdot)$ sorts indices in ascending order. The KV pairs corresponding to other non-multimodal tokens, $\mathbf{I}_{\text{other}}$, are added back, maintaining their relative positions unchanged, which is crucial to ensure the correct functioning of the model.

After $\lceil N/m \rceil$ iterations, all remaining KV pairs contain highly condensed content that is relevant to the current question. The LLM then generates a response based on the prompt and the extracted KV pairs. This mimics the process of extracting task-specific information from long-term memory to working memory in the human brain (Jeneson & Squire, 2012; Chai et al., 2018). Since the memory size is fixed, it still satisfies streaming constraints.

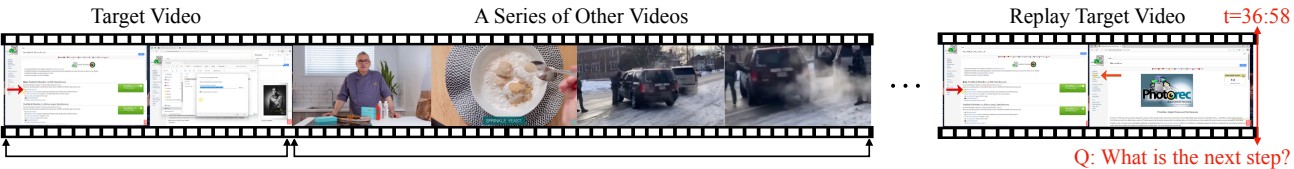

*Figure 4.* Illustration of the ELViM data. The question is given at a specific timestamp, asking about the next step to execute. The answer can be found in the target video, which has been watched for at least 30 minutes before the time of the question.

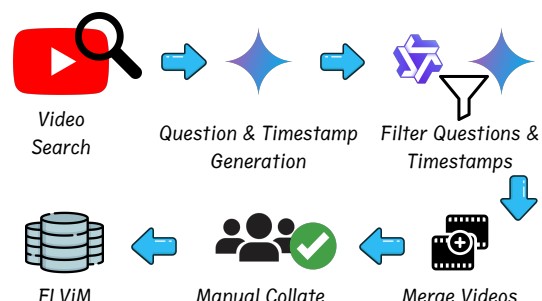

*Figure 5.* Data creation and annotation pipeline for ELViM.

## 4. Episodic Learning from Video Memory

Humans often acquire skills by observing demonstrations, increasingly through short videos, and recall these skills later from long-term memory when performing similar tasks. The ELViM benchmark is designed to simulate this process by evaluating whether LLMs can learn procedural knowledge from episodic video memory and retrieve it to support real-time task execution. As illustrated in Fig. 4, a target video containing procedural information is first presented, followed by a long sequence of unrelated videos. After an extended interval, the target video is replayed and paused at a specific moment just before the demonstrator performs the next step, at which point the model is queried about the upcoming action.

ELViM differs from existing knowledge acquisition benchmarks such as VideoMMMU (Hu et al., 2025) in two key aspects: (i) the videos are substantially longer, lasting up to 2 hours, with the relevant knowledge appearing at least 30 minutes before the query; and (ii) questions are posed at a precise time point within an ongoing video, requiring real-time awareness of task progress. As a result, ELViM tests real-time perception and long-term memory, reflecting the requirements of future real-world agents that must acquire and reuse skills from extended audio-visual experiences.

### 4.1. Data Creation Pipeline

We adopt a rigorous data creation pipeline to avoid knowledge leakage or textual shortcuts, as shown in Fig. 5.

**Video collection and question generation**: We first collect over 1000 tutorial or demonstration videos spanning 15 categories that were uploaded after mid-2025. We then generate 10 questions about certain procedures for each

video, together with their corresponding timestamps and distracting choices using Gemini-2.5-Pro.

**Question and timestamp filtering**: Next, we ensure the questions cannot be answered by the model's own knowledge or inference from previous steps. We prompt Gemini-2.5-Pro and Qwen3-VL-8B with questions and video up to the timestamp when the target procedure begins. We then kept the questions that those models fail to answer with incomplete videos, but can answer with complete ones.

**Rest of the steps**: At last, we concatenate single videos into long merged videos, each up to 90 minutes, where the target video is at least 30 minutes away from the question timestamp. Thereafter, human collation is performed for the correctness of the question and the accuracy of the timestamp in the merged video. We provide detailed statistics of the ELViM dataset as well as the detailed performance on single videos in Appendix A. Example questions and answers are provided in Appendix B.

## 5. Experimental Setup

### 5.1. Model

video-SALMONN S is built based on the Qwen3-VL 8B backbone (Bai et al., 2025a). Following video-SALMONN-2 (Tang et al., 2025a), Whisper-Large-v3 encoder (Radford et al., 2023) is used to encode audio information and a window-level Q-Former with a window length of 0.5 seconds as the audio aligner. The LLM backbone is trained with LoRA using a rank of 128. Without memory reading, we experiment with 16k memory tokens with 1k tokens allocated for real-time perception. With memory reading, we double the amount of memory tokens with a chunk size of 8k, so that the GPU memory is similar to 16k tokens.

$\text{TTT}_{\text{MEM}}$ contains 2 fully-connected layers with GeLU activation function. The input encoding is converted into 8 heads, where each head is 512-dimensional and is handled by one fast weight, yielding sixteen $512 \times 512$ fast weights. The trainable learning rate $\eta$ follows the same setting as (Dalal et al., 2025). We use stochastic gradient descent with one update step for $\text{TTT}_{\text{MEM}}$. We set $T = 2$ as the default for long-span prediction, and experimented with $T$ ranging from 1 to 32 as shown in Table 10 in Appendix F. Detailed FLOP statistics of $\text{TTT}_{\text{MEM}}$ and base models are provided in

Appendix D. As a design choice for Qwen3-VL, deepstack embeddings are directly discarded according to the indices at the output of $TTT_{MEM}$ without additional $TTT_{MEM}$, based on the results in Appendix E.

## 5.2. Training Configurations

Following (Tang et al., 2025a), the audio aligner of video-SALMONN S is first trained using LibriSpeech 960-hour (Panayotov et al., 2015), CommonVoice (Ardila et al., 2020), WavCaps (Mei et al., 2024), and AudioCaps (Kim et al., 2019), with other parts of the LLM frozen. Then, FineVideo (Farré et al., 2024), CinePile (Rawal et al., 2024), and about 100k videos sampled from LLaVA-Video-178k are used to finetune the entire audio-visual model following the two-stage training scheme.

During stage 1 training, we use a 4 FPS sampling rate with a maximum of 1024 frames in total to achieve a better trade-off between parallelisation and sequence lengths. The first stage trains for 3 epochs with a learning rate of $2\times10^{-5}$, which takes 48 hours on 32×H800 GPUs. In stage 2 training, we increase the number of frames to 2048 and train for another epoch, which takes 16 hours on 32×H800 GPUs. We use 1 sample per minibatch on one GPU.

## 5.3. Evaluation Configurations

Following recent studies of memory in streaming models (Yang et al., 2025a; Zeng et al., 2025), we evaluate video-SALMONN S on both general long video benchmarks, where the system operates in streaming mode, as well as streaming-specific benchmarks with long videos.

**Benchmarks**: For long video benchmarks, we use **Video-MME** (Fu et al., 2024) which contains several minutes to 1 hour videos. We also include **LVBench** and **VideoEvalPro** as two extremely long video QA benchmarks. LVBench (Wang et al., 2024b) contains videos of several hours that focus on long-term memory and diverse core capabilities, and VideoEvalPro (Ma et al., 2025) also contains videos of several hours, and is designed to more realistically evaluate long video understanding without MCQ shortcuts, where the MCQ partition is used. For streaming-specific benchmarks, we primarily compare systems on our proposed **ELViM** benchmark, and at the same time evaluate on **Streaming-Bench** (Lin et al., 2024a). Note that StreamingBench does not contain long videos where long-term memory is necessary. The experiments are mainly to demonstrate the state-of-the-art performance of video-SALMONN S on standard online video tasks. Percentage accuracies are reported on all these benchmarks.

**Baselines**: We employ 2 streaming baselines for our experiments, including memory consolidation (Song et al., 2024; Chen et al., 2024) and Persistent event memory forest (PEMF) in StreamForest (Zeng et al., 2025)[2]. We implement the memory algorithms in each of the baselines in our model to ensure they have the same backbone LLM and the same training data. We also compare with strong non-streaming baselines, including Qwen3-VL itself and video-SALMONN 2 (Tang et al., 2025a). Regarding the reading mechanism, we compare with AdaReTaKe (Wang et al., 2025b) on the same model.

For streaming models, the video is processed at 1 FPS by default, and has an alternative 4 FPS for videos less than 2 minutes (*i.e.* exclusively used for Video-MME short partition). For non-streaming models, we use up to 10 FPS sampling rate and apply a maximum token limit of 16k, and uniformly sample frames when exceeding the limit.[3]

## 6. Results

### 6.1. Main Results

We first present the main results in Table 1, where video-SALMONN S operates under an effective memory budget of 16k memory tokens corresponding to 215 input video frames for non-streaming models, and $\sim$ 24GB GPU memory, respectively. Note that all systems in Table 1 are based on the same LLM backbone and the same training data, and hence they only differ in how the long-term memory is constructed. For PEMF, we use the default penalty coefficients (0.4, 0.4, 0.2) from the original paper.

Overall, **video-SALMONN achieved the best performance** on standard long video benchmarks compared to both streaming and non-streaming baselines. Specifically, with 16k memory tokens compared to its non-streaming counterpart, video-SALMONN S achieved 1.7%, 2.9% and 5.4% accuracy improvements with 16k memory tokens on Video-MME long, LVBench and VideoEvalPro, respectively. This advantage further increases compared to the streaming counterparts, with 1.9%, 4.0% and 7.1% absolute accuracy improvements. Such large gains achieved by video-SALMONN S demonstrate its superior long-term memory design. On StreamingBench, where long-term memory is not critical, video-SALMONN S still achieves competitive performance in real-time perception tasks.

With 80GB memory GPUs, we could further increase the number of memory tokens, and we provide results when using 128k memory in Appendix **??** to show the extreme case on a single GPU. Note that with this memory size, videos less than 30 minutes at 1 FPS do not need any merging or token compression, and hence we should focus on the results on longer benchmarks.

---

[2]Due to the large number of visual tokens, we compute penalty and perform merging on-the-fly for every 64 incoming frames

[3]Code implementation of video-SALMONN S and ELViM questions are provided in the supplementary material.

*Table 1.* Performance of video-SALMONN S on various benchmarks compared against baselines. Numbers are all percentage accuracies. The short, medium and long partitions (S/M/L) are reported separately for Video-MME, and the real-time, omni-modal and contextual partitions (R/O/C) are reported for StreamingBench. Note that all baseline methods are implemented using the same Qwen3-VL base and the same training data, which are all *better* than the reported numbers in original papers.

| Model | Video-MME (S/M/L) | LVBench | VideoEvalPro | ELViM | StreamingBench (R/O/C) |
|---|---|---|---|---|---|
| Effective memory budget 16k memory tokens | | | | | |
| Qwen3-VL 8B (Bai et al., 2025a) | 69.7 (80.7/68.3/60.1) | 47.4 | 54.9 | 28.1 | 61.8 (76.9/42.5/37.5) |
| video-SALMONN 2+ (Tang et al., 2025a) | 75.6 (81.6/75.7/69.6) | 52.7 | 53.5 | 32.5 | 66.4 (77.7/57.5/41.0) |
| Similarity Merging (Song et al., 2024) | 75.8 (82.0/76.2/69.4) | 51.6 | 51.8 | 38.5 | 66.8 (78.1/59.5/39.2) |
| PEMF (Zeng et al., 2025) | 75.8 (82.7/76.1/68.6) | 49.5 | 50.4 | 38.2 | 67.1 (79.1/57.1/40.8) |
| AdaReTaKe (Wang et al., 2025b) | 76.1 (81.9/76.3/70.2) | 54.3 | 54.5 | 41.5 | 67.0 (78.9/57.0/40.8) |
| video-SALMONN S (w/o reading) | 76.4 (82.7/75.2/71.3) | 55.4 | 55.8 | 43.9 | 66.8 (78.4/57.5/40.9) |
| video-SALMONN S (w reading) | **76.9** (82.5/76.8/71.3) | **55.6** | **58.9** | **46.7** | **67.1** (78.9/57.5/41.5) |

Moreover, **video-SALMONN S achieved significantly better performance on ELViM** compared to baselines. For non-streaming baselines, sparse frame sampling may omit the target video in history, as well as omit the real-time perception of the current progress, therefore leading to the worst performance overall. Similarity merging and PEMF, on the contrary, maintain good real-time perception abilities and hence perform better than the non-streaming baseline. However, both streaming methods suffer from long-term memory, so even if they correctly perceive the progress, they struggle to find the correct answer to it.

This is also reflected by the performance differences of PEMF and similarity merging on StreamingBench and long video benchmarks. Our observation on Video-MME and StreamingBench generally aligns with the original paper (Zeng et al., 2025). While similarity merging and PEMF perform well on short to medium-length videos, their performances degrade on videos of 3-4 hours in length. The differences between PEMF and similarity merging on LVBench and VideoEvalPro are mainly due to the temporal penalty. In contrast, $TTT_{MEM}$ achieves a better balance by incorporating both near and far history information into fast weights. As a result, video-SALMONN S achieves 14.2% improvements compared to the non-streaming baseline, and 8.8% to PEMF with 16k memory tokens. Qualitative case studies are provided in Appendix C to further support our claims.

We also provide the performance of video-SALMONN S when larger memory is available, from 32k memory tokens up to 128k memory tokens, as shown in Table 2.

## 6.2. Ablation Studies

We provide **the contribution of different design choices in video-SALMONN S** in Table 3. First, we compare different downsampling techniques, including merging, clustering and discarding. We observe that merging and discarding achieved similar performance, that are both superior to clustering of tokens. Then, we compare the effect of $TTT_{MEM}$

against TTT-video[4] and Mamba 2 as two alternative RNN-type long-term memory mechanisms. We observe that TTT-video achieved consistently better performance than Mamba-2 in long video understanding. $TTT_{MEM}$ further improves the performance of TTT-video by 1-2%, which is mainly due to the newly introduced long-span prediction loss. The second stage training further improves long video understanding by 0.2-0.4%, with a modest effect on relatively short videos such as Video-MME long.

The comparison between AdaReTaKe (Wang et al., 2025b), a recent training-free KV-Cache compression method, and our proposed reading mechanism is also performed. As shown in Table 3, our modality-aware reading mechanism achieved consistent and significant performance improvements compared to AdaReTaKe.

**Influence of memory size**: The variation of long-term memory performance on ELViM and LVBench against different memory sizes is shown in Fig. 6. Note that memory reading is not used, so that the comparison is purely focused on the two long-term memory mechanisms. Overall, $TTT_{MEM}$ shows a more robust trend than Merging when the number of memory tokens reduces to 4k and 2k, and maintains its advantage across all different memory sizes we evaluated. Notably, on both datasets, $TTT_{MEM}$ achieves similar accuracies as merging baselines with only equal to or less than **25%** of memory tokens used by merging, demonstrating its strong long-term memory capability.

**Influence of number of frames**: The performance change against increasing number of input frames on ELViM and LVBench is shown in Fig. 7, with 16k memory token budget for both methods. Again, the reading mechanism is excluded to focus on long-term memory mechanisms. While both methods show an upward trend when the number of frames increases from 256 to 2048, the similarity merging method suffers significantly when the number of frames continues to grow exponentially. On the contrary, $TTT_{MEM}$

---

[4]TTT-video refers to the standard TTT in (Dalal et al., 2025).

*Table 2.* Performance when further increasing the number of memory tokens. 128k tokens is the upper limit on a single H800 GPU.

| Setup | Video-MME | LVBench | VideoEval-Pro | ELViM |
|---|---|---|---|---|
| 32k Mem w/o reading | 77.1 (82.9/76.3/72.0) | 57.7 | 59.2 | 47.1 |
| 32k Mem w/ reading | 77.3 (82.5/77.2/72.1) | 57.9 | 61.0 | 49.4 |
| 128k Mem w/o reading | 77.4 (82.8/77.1/72.4) | 59.6 | 62.0 | 53.2 |
| 128k Mem w/ reading | 77.3 (82.7/77.1/72.1) | 59.8 | 62.3 | 54.9 |

*Table 3.* Ablation studies on different components in video-SALMONN S with 16k memory token settings on long video benchmarks without memory reading. "Reading" refers to our proposed modality-aware reading mechanism.

| Configuration | Video-MME Long | LVBench | VideoEvalPro | ELViM |
|---|---|---|---|---|
| Similarity Merging (Song et al., 2024) | 69.4 | 51.6 | 51.8 | 35.4 |
| K-means Clustering (Zhou et al., 2024) | 67.8 | 49.8 | 50.7 | 33.8 |
| Similarity Discarding | 69.2 | 51.1 | 51.5 | 36.8 |
|   + Mamba-2 (Dao & Gu, 2024) | 70.3 | 53.5 | 54.8 | 39.6 |
|   + TTT-video (Dalal et al., 2025) | 70.0 | 54.3 | 54.9 | 42.3 |
|   + $TTT_{MEM}$ w/o Stage 2 Training | 71.3 | 55.1 | 55.5 | 43.6 |
|   + Stage 2 Training (video-SALMONN S w/o reading) | 71.3 | 55.4 | 55.8 | 43.9 |
| video-SALMONN S + AdaReTaKe (Wang et al., 2025b) | 70.8 | 55.1 | 56.5 | 43.6 |
| video-SALMONN S + Reading | **71.3** | **55.6** | **58.9** | **46.7** |

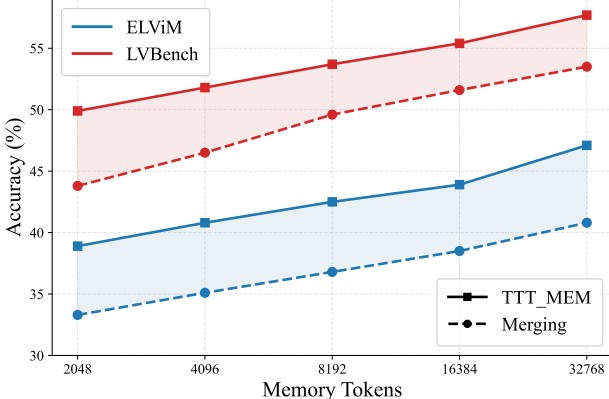

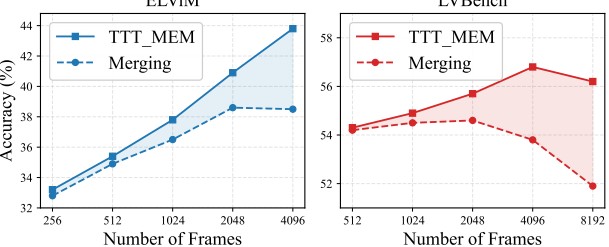

*Figure 7.* Variation of model performance against different numbers of input frames on ELViM and LVBench using similarity merging and $TTT_{MEM}$ without the reading mechanism. Both methods use 16k memory tokens. The X-axis is on a log scale.

*Figure 6.* Model performance against different numbers of memory tokens on ELViM and LVBench using merging and $TTT_{MEM}$ without reading mechanism. $TTT_{MEM}$ achieves the same level of accuracy with less than 25% of memory tokens needed by merging.

shows a steady upward trend up to 4096 frames, and reaches the plateau after that without obvious degradation. The onset of the plateau is determined by the number of memory tokens budget, and can be significantly postponed to over 10k frames using 32k memory tokens.

**Influence of long-span prediction**: Since LVBench also provides the time at which the answer could be found in the video, we plot the model performance against the temporal distance between the evidence location and the query time in Fig. 8. We compare $TTT_{MEM}$ with and without long-span prediction in this plot to demonstrate that the long-span prediction helps to retain information. Long-span prediction yields consistent improvements as the temporal distance

between the query and evidence increases, demonstrating its effectiveness in modeling long-term dependencies.

### 6.3. Runtime Statistics

*Table 4.* Memory and inference time per sample measured for Video-MME long at 1 FPS (maximum 3600 frames) on a single H800 GPU with 16k memory tokens.

| Model | GPU Mem. (Avg./Peak) | Infer. Time (second) |
|---|---|---|
| Similarity Merging | 21.9 / 24.1 | 10.4 |
| PEMF | 21.9 / 24.1 | 10.4 |
| v-SALMONN S (w/o Reading) | 22.2 / 24.3 | 10.5 |
| v-SALMONN S (w/ Reading) | 22.5 / 24.4 | 10.9 |

We compare runtime statistics among different streaming methods as shown in Table 4, with the same number of input frames. All models consumes similar amount of GPU

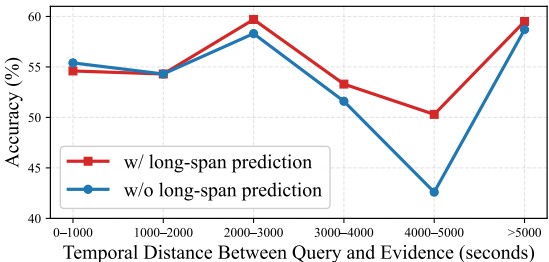

*Figure 8.* Comparison of video-SALMONN S with and without long-span prediction on LVBench at different temporal distances between the query and the video evidence.

memory. The inference time is the time taken from inputting the video to generating the entire answer, which is around 10 seconds. Most of the time is spent on encoding visual inputs, as there are far more frames to deal with than short videos. However, the average time taken for each frame is very short, and this can be done on-the-fly. $TTT_{MEM}$ only creates 0.1s overhead on top of this. Reading mechanism costs another 0.4s due to the iterative prefill of token chunks into KV-Cache in Transformer.

## 7. Conclusion

In this paper, we propose video-SALMONN S, a memory-enhanced video LLM that processes over 3-hour videos at 1 FPS and 360p resolution. video-SALMONN S is the first to explore test-time training in long-term memory modeling. We include novel components, including a long-span prediction objective, a two-stage training pipeline and a modality-aware memory reading mechanism. In addition, we propose an episodic learning from video memory (ELViM) benchmark, which tests whether LLMs can learn from past experiences and apply them to real-time tasks. Consistent and significant improvements compared to both streaming and non-streaming baselines were achieved on a range of long video benchmarks. Notably, a 14.2% absolute accuracy improvement was achieved on ELViM.

## Impact Statement

This work introduces video-SALMONN-S, a memory-enhanced streaming audio-visual large language model designed to process long-duration videos under fixed memory constraints. By enabling models to consolidate information from hours-long video streams and recall it when needed, this research advances the development of AI systems that can learn from extended experience, a capability that is important for future assistive agents, educational tools, and long-horizon decision-making systems.

The proposed approach may benefit applications such as instructional assistance, long-form educational video understanding, human–computer interaction, and support for

users who rely on audio-visual information over extended periods (e.g., accessibility tools for visually or cognitively impaired users). The introduced ELViM benchmark may also encourage more rigorous evaluation of long-term memory and continual learning in multimodal systems, contributing to more transparent and realistic assessment practices in the research community.

Models capable of long-term memory accumulation from video streams could raise privacy concerns if deployed on sensitive or personal video data without appropriate safeguards. There is also a risk that long-term memory mechanisms may inadvertently encode biased, misleading, or harmful content observed in training or deployment environments, potentially amplifying such information over time. Additionally, the computational and energy costs associated with large multimodal models may contribute to environmental impacts if deployed at scale.

These risks can be mitigated through careful dataset curation, deployment-time privacy controls, explicit user consent for long-term memory storage, and mechanisms for memory inspection, reset, or forgetting. Bias and safety auditing should be applied to both training data and downstream deployments, particularly in agent-like scenarios. From a systems perspective, continued work on efficiency and memory-constrained operation, as explored in this paper, can help reduce unnecessary computational overhead.

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

## A. ELViM Detailed Statistics

We provide descriptive statistics of ELViM in Table 5, including the number of questions, the number of videos and duration statistics of the merged videos. We also provide the distribution of categories used ELViM in Table 6. Note that some of the search queries are generic, but the actual tutorials or procedures differ greatly from video to video.

*Table 5.* Statistics of ELViM dataset.

| Number of Questions | Number of Target Videos | Durations (Average/Max/Min) |
| --- | --- | --- |
| 1849 | 1021 | 2920s / 5518s / 1822s |

*Table 6.* Category distribution of ELViM dataset.

| Category | Number of target videos | Example search queries |
| --- | --- | --- |
| Home repair | 68 | How to replace a showerhead without leaks
Replace window screen mesh walkthrough |
| Car & bike maintenance | 59 | How to align bike brakes tutorial
Flush car coolant step by step |
| Cooking fundamentals | 94 | How to sharpen kitchen knives with stone
Scrambled eggs creamy technique tutorial |
| Baking & desserts | 80 | How to make choux pastry beginner guide
Why cookies spread too much, troubleshooting |
| Tech help | 124 | Bluetooth connection issues troubleshooting
Phishing email identification guide |
| Software skills | 45 | Notion task management setup guide
Photoshop crop and straighten tutorial |
| Money & admin life skills | 56 | Understand insurance deductible explained
How to choose a savings account |
| Cleaning & home organization | 84 | Spring cleaning whole house checklist
Organize paperwork at home |
| Sewing & clothing repair | 42 | Adjust waistband size tutorial
Hand sew invisible stitch tutorial |
| DIY crafts & maker skills | 67 | 3D printing filament types explained
Hand tool woodworking starter skills |
| Gardening & plants | 64 | Prune houseplants beginner guide
How to stake plants properly |
| Outdoors & practical skills | 67 | Pack a backpack efficiently tutorial
Identify animal tracks beginner guide |
| Communication & career skills | 54 | Set boundaries at work tutorial
How to say no professionally |
| Fitness skills | 70 | Proper breathing during exercise
Balance exercises for beginners |
| Art & creative skills | 47 | Gesture drawing beginner exercises
Camera movement techniques explained |

The data creation pipeline guarantees that the videos are not seen in the training set by Gemini-2.5-Pro and Qwen3-VL. As shown in Table 7, at the time when the question is asked, without having watched the video, both Qwen3-VL and Gemini-2.5-Pro produced an accuracy that is close to random guessing. With the entire video provided, both models can achieve an accuracy close to 100%, up to some visual perception errors.

*Table 7.* Performance of Gemini-2.5-Pro and Qwen3-VL 8B on ELViM questions with single videos.

| Model | Setup | Accuracy |
|---|---|---|
| Qwen3-VL 8B | Video up to question timestamp | 22.6% |
| Qwen3-VL 8B | Full video | 98.7% |
| Gemini-2.5-Pro | Video up to question timestamp | 23.5% |
| Gemini-2.5-Pro | Full video | 99.3% |

## B. ELViM Examples

We provide six example questions of ELViM as shown in Fig. 9. The first example is a video demonstrating how to use a specific software, and about 1 second after the last frame shown, the demonstrator moves the cursor to the left menu to download the software. The second video demonstrates how to cook something. The question asks about the specific requirements of the machine shown in the white box in the last couple of frames. This is explained immediately in 2-3 seconds by the demonstrator. This question requires the model to understand what it means by "the machine" that appears at the moment, and associate with its memory about what the requirements for the machine are. The rest of the examples follow a similar design to the above two types.

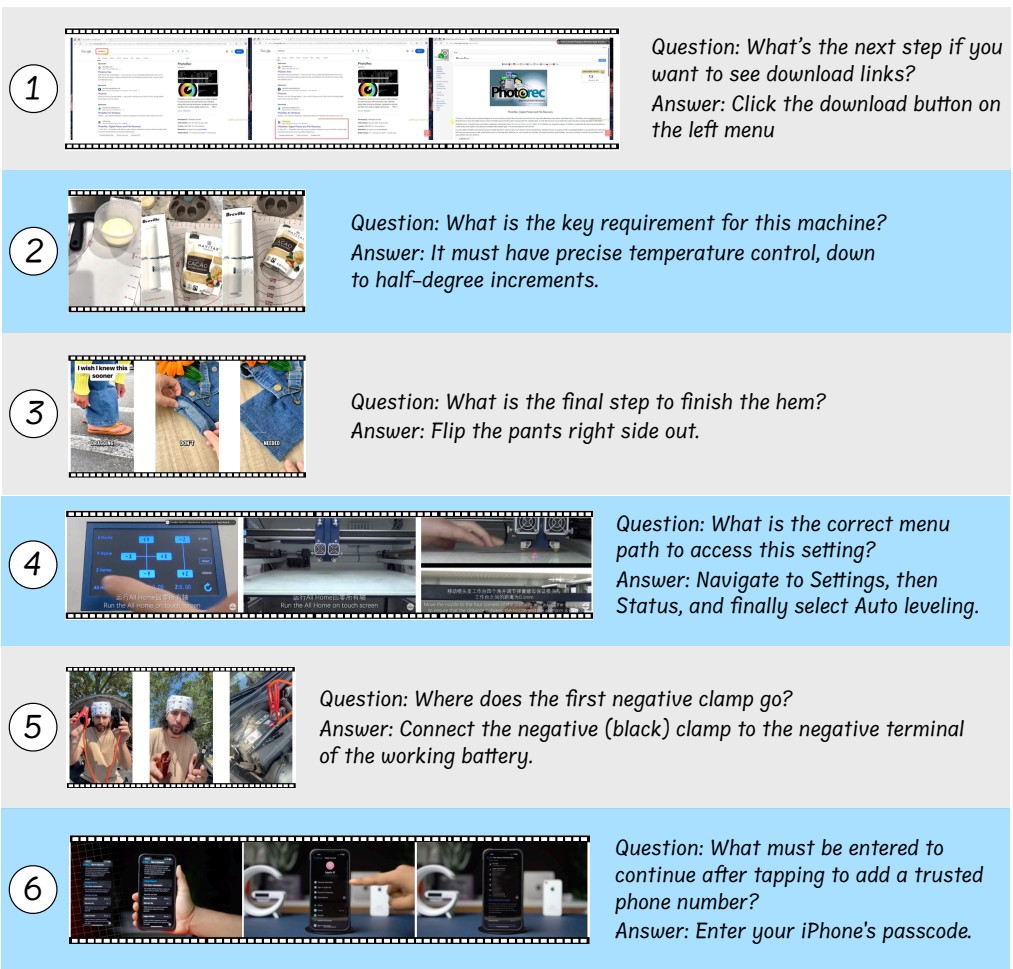

*Figure 9.* Examples of questions and answers in ELViM. The categories of these examples are: ① Software skills; ② Baking and dessert; ③ Sewing and clothing repair; ④ Tech help; ⑤ DIY crafts and maker skills; ⑥ Money and admin life skills.

# C. Qualitative Analysis on ELViM

We provide case studies on ELViM and compare the responses from different models to questions in ELViM. We convert the question into open-ended questions and let the model generate freely without giving it choices. Three examples are provided in Fig. 10 to 12 respectively.

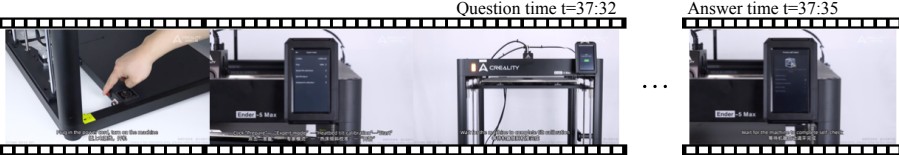

Question: What is the recommended next action after calibration?

Reference: Perform auto-leveling and wait for self-check to complete

Non–streaming: Checking calibration results

PEMF: Press the button on the screen for the next step

video–SALMONN S: Press the green button and wait for self-check to complete

*Figure 10.* Case study on non-streaming model, PEMF and video-SALMONN S in response to questions from ELViM. In this example, the non-streaming model fails to understand the current status of the video and hence makes an educated guess based on the given question.

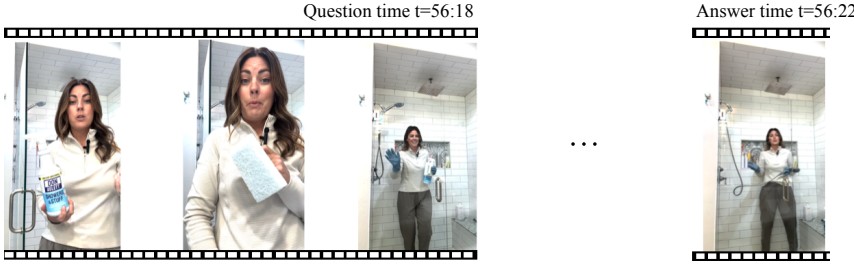

Question: What is a crucial next step before applying the cleaning product to the glass?

Reference: Pre-rinse the entire glass with cold water.

Non–streaming: Wetting the surface with cold water.

PEMF: Spray the cleaning solution onto the glass door and then use a scraper to gently lift the top layer of the hard water stains.

video–SALMONN S: Wetting the glass with cold water.

*Figure 11.* Case study on non-streaming model, PEMF and video-SALMONN S in response to questions from ELViM. In this example, the woman has been holding the spray and a scraper in recent frames, and PEMF hallucinates that the woman is going to spray and scrape.

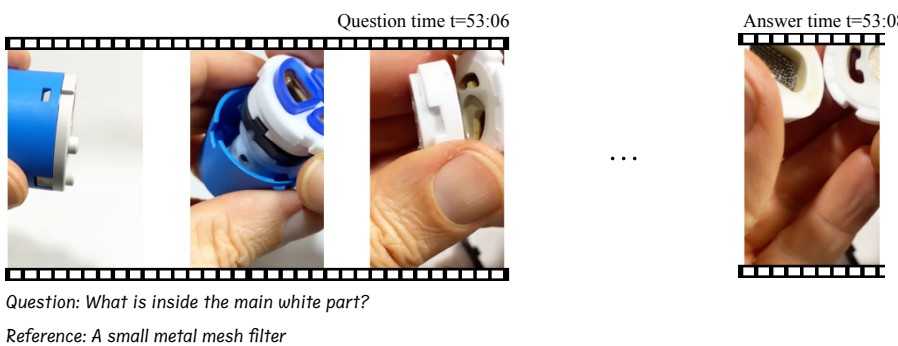

Question: What is inside the main white part?

Reference: A small metal mesh filter

Non–streaming: a ceramic disc and a red plastic component

PEMF: A small metal pin in the center

v–SALMONN S: A mesh filter

*Figure 12.* Case study on non-streaming model, PEMF and video-SALMONN S in response to questions from ELViM. In this video, both PEMF and non-streaming baseline fail to accurately identify the *mesh* structure of the internal.

## D. FLOP Statistics

FLOP statistics are provided in Table 8. Incorporation of the $TTT_{MEM}$ layer only adds a negligible overhead to TFLOPs compared to the entire Transformer.

*Table 8.* TFLOPs for video-SALMONN S

| Configuration | TFLOPS |
|---|---|
| Qwen3-VL 8B | 3.15 |
| video-SALMONN S (w/o reading) | 3.15+0.000235 |

## E. Ablation Studies on Deepstack Embeddings

We provide the results in Table 9 to justify our design choice that only incorporates the $TTT_{MEM}$ layer at the Transformer input, and performs direct downsampling/discarding to the deepstack tokens. Including $TTT_{MEM}$ for each stream of deepstack embeddings will multiply the additional memory and runtime overhead by at least 4, and also significantly increase training memory consumption.

*Table 9.* Accuracies on long-video benchmarks with $TTT_{MEM}$ applied to only Transformer inputs and $TTT_{MEM}$ applied to all 4 layers with deepstack embeddings for Qwen3-VL

| Model | VideoMME (Short/Medium/Long) | LVBench |
|---|---|---|
| Inputs Only | 76.4 (82.7/75.2/71.3) | 55.4 |
| Inputs + Deepstack | 76.1 (82.2/76.0/70.1) | 54.7 |

## F. Ablation Studies on the Prediction Spans

We show the accuracy against different prediction span settings in Fig. 10. All spans achieved equal or better performance compared to TTT without prediction. However, a longer span produces much noisier information and becomes less effective, and spans within one TTT minibatch do not have a large influence as well. Therefore, we choose 2 as the best trade-off.

*Table 10.* Performance of TTT-video and $TTT_{MEM}$ with different prediction spans, T, on Video-MME long partition and LVBench with 16k memory tokens.

| Span | Number of Tokens Intervals | Video-MME long | LVBench |
|---|---|---|---|
| TTT-video | – | 70.0 | 54.3 |
| T=1 | 1024 | 70.6 | 54.5 |
| T=2 | 2048 | **71.3** | **55.4** |
| T=4 | 4096 | 70.9 | 55.1 |
| T=16 | 16384 | 71.0 | 54.8 |
| T=32 | 32768 | 70.9 | 55.0 |

