# OpenReview forum: "video-SALMONN S: Memory-Enhanced Streaming Audio-Visual LLM"
_ICML.cc/2026/Conference — ICML 2026 regular_

### Official Review · Reviewer_9g86 · 2026-03-04

**Soundness:** 3
**Presentation:** 2
**Significance:** 3
**Originality:** 3
**Overall Recommendation:** 4
**Confidence:** 4

**Summary:**

This paper proposes video-SALMONN S, a streaming video understanding model with an enhanced memory mechanism. To address information loss over long durations in existing models, the paper adopts test-time training (TTT) to preserve long-term memory. A TTT memory layer is introduced to retain long-term information within a fixed memory budget, with token discarding and prompt-dependent memory reading to further improve performance. The authors also introduce a benchmark called ELViM, which aims to evaluate a model’s ability to acquire knowledge from hour-scale episodic videos. Results show that the proposed model achieves strong performance on both streaming and non-streaming video understanding benchmarks, including the proposed ELViM benchmark.

**Compliance With Llm Reviewing Policy:**

Affirmed.

**Final Justification:**

Most of my concerns were well addressed during the rebuttal stage. The authors demonstrated the effectiveness of using TTT for streaming understanding, and the proposed benchmark is likely to be a useful resource for the community.

That said, the presentation of the paper still needs improvement, particularly with respect to the dataset construction details and several issues in the current writing. For these reasons, I choose to maintain my original score.

**Key Questions For Authors:**

1. What is the difference between ELViM with existing Benchmarks?
2. Will ELViM be publicly released?
3. Why do the authors still include Stage 2 training?

**Limitations:**

yes

**Strengths And Weaknesses:**

**Strengths**:

1. The use of TTT in streaming video understanding has not been thoroughly explored in prior work. The motivation is clear, and the method design is intuitive and well-justified. This work provides a strong baseline that can be used for future exploration.

2. The proposed ELViM benchmark covers diverse domains and provides a more challenging benchmark for future research.

3. The implementation details are clearly presented, and the authors also provide code, which benefits reproducibility for the research community.

4. The model achieves strong performance over several important baselines and also demonstrates clear memory efficiency.

**Weakness**:

1. ELViM serves as a core contribution of this paper, but the motivation for constructing this benchmark is not sufficiently clear. The differences between ELViM and existing long video understanding benchmarks (e.g., EgoScheme and StreamingBench) are not fully discussed, such as covered scenarios, duration distributions, and the number of events. A detailed statistical analysis and comparison with existing benchmarks should be added to clarify what is novel in ELViM.

2. TTT is the core module of the model. However, the effect of the number of parameters on performance is neither analyzed nor discussed.

3. Some results are not fully discussed. In Table 2, Stage 2 training yields very limited performance gains. Given the training cost (16 hours on 32×H800), why do the authors still include Stage 2 training?

4. Some notations are introduced without clear definitions: $T$ is not formally defined when first mentioned (Line 149), and $\theta_{K}$, $\theta_{V}$, and $\theta^{'}_{K}$ are also not clearly defined.

---

> ### Author Rebuttal · Authors · 2026-03-30
>
> We appreciate the encouraging positive feedback and insightful questions from the reviewer. We would like to address each individual concerns as follows:
>
> > Weakness 1 & Question 1: ELViM serves as a core contribution of this paper, but the motivation...what is novel in ELViM. What is the difference between ELViM with existing Benchmarks?
>
> To our knowledge, ELViM is the __first benchmark__ that tests whether an LLM can acquire knowledge from __a few-hour video memory__ and apply it to __real-time__ tasks:
> - Compared to EgoScheme and other benchmarks longer than 1 hour: Videos in ELViM are stopped at a specific time and the LLM needs to __answer what to do next at this moment__.
> - Compared to StreamingBench and other streaming benchmarks: ELViM requires memory __spanning multiple hours__, and memory is __indispensable__ to answer the question.
> - Moreover, ELViM focuses on __procedural knowledge__, which is largely overlooked by long video benchmarks.
>
> > Weakness 2: TTT is the core module of the model. However, the effect of the number of parameters on performance is neither analyzed nor discussed.
>
> TTT introduces negligible number of parameters compared to LLM:
> - Total number of trainable parameters in TTT is around 80M (around __10%__ of LoRA parameters and __1%__ of total parameters).
> - We found that increasing LoRA or the aligner layer by 10% instead of using TTT __does not yield any significant improvements__.
> - Appendix D shows that TTTMEM adds __only 0.000235 TFLOPs__ on top of a __3.15 TFLOP__ Qwen3-VL 8B forward pass (0.0075%).
>
> > Weakness 3 & Question 3: Some results are not fully discussed. In Table 2, Stage 2 training yields very limited performance gains. Given the training cost (16 hours on 32×H800), why do the authors still include Stage 2 training? Why do the authors still include Stage 2 training?
>
> - The __improvement from stage 2 is consistent across long video benchmarks__, including LVBench, VidEvalPro, and ELViM, which further advances performance.
> - Stage 2 is __necessary__ allows us to use a higher memory budget. For example, when 32k memory tokens can be used (e.g. for the results in Table 9), we need to perform stage 2 to adapt the model that was previously trained with 16k memory tokens to 32k.
>
> We will make this clearer in the revised paper.
>
> > Weakness 4: Some notations are introduced without clear definitions...are also not clearly defined.
>
> We appreciate the careful reading of the reviewer, and we will include the following changes:
> - $T$ is the number of minibatches before the current step $t$ that we predict for the long-span loss.
> - We enhance our description after Eqn. (3): $\theta_{QKV} \in \mathbb{R}^{D_K \times (HD_W)}$ are trainable projection matrices that project inputs $X$ into the space of W, where $D_K$ is the dimension of key vectors, $D_W$ is the dimension of fast weights, and $H$ is the number of heads.
>
> > Question 2: Will ELViM be publicly released?
>
> Yes, of course. The questions and YouTube IDs of ELViM have already been included in the supplementary materials.

---

> > ### Author Rebuttal · Reviewer_9g86 · 2026-04-01
> >
> > Thanks for your response. Most of my concerns are addressed.
> >
> > I am still confusing about the effect of stage 2 training. You mentioned that "improvement from stage 2 is consistent across long video benchmarks", but from line 393 and 394 we can see that the improvement is limited (less than 0.5 for all dataset). Could you explain that?

---

> > > ### Author Response · Authors · 2026-04-02
> > >
> > > Thank you for your insightful follow-up. We agree that the improvement from Stage 2 under a __fixed__ memory budget is modest (<0.5). We will refine our language to ensure the distinction between __performance gain__ and __capacity adaptation__ is clear.
> > >
> > > The key point is that Stage 2 is not intended to provide a large standalone gain under an unchanged configuration, but to __enable the model to operate effectively under a larger memory budget__. As described in Lines 191–196, Stage 2 serves two purposes:\
> > > (1) increasing the number of input video frames during training, and\
> > > (2) increasing the memory budget during training.
> > >
> > > We agree that the gain from __Purpose (1)__ at a fixed memory budget is modest. This is expected because most training videos are around 10 minutes long, so increasing the input length from 1024 to 2048 frames does not substantially change the effective supervision. That said, the gain is still directionally consistent: a two-tailed sign test at the per-video level across the three benchmarks yields __p = 0.04__ (36 positive vs. 20 negative cases), suggesting the improvement is small but statistically reliable rather than noise.
> > >
> > > However, the more important role of Stage 2 is __Purpose (2): adapting the model to a larger memory budget__. This is the main practical reason why Stage 2 is needed. For example, when both models are evaluated with a __32k memory budget__ at inference time, the Stage 2 model is substantially better than the Stage 1 model:
> > > | Model | VMME long | LVBench | VidEvalPro | ELViM |
> > > |-------|-----------|---------|------------|-------|
> > > | Stage 1 | 71.5 | 55.9 | 56.3 | 44.3 |
> > > | Stage 2 | __72.0__ | __57.7__ | __59.2__ | __47.1__ |
> > >
> > > Therefore, one key value of Stage 2 is __capacity adaptation__: it allows a model trained at the Stage-1 setting to function effectively at a higher memory budget, such as 32k. We will revise the paper to make this distinction explicit, and we will __move Table 9 to the main text__ together with this comparison.

---

### Official Review · Reviewer_959h · 2026-03-11

**Soundness:** 3
**Presentation:** 3
**Significance:** 3
**Originality:** 3
**Overall Recommendation:** 4
**Confidence:** 4

**Summary:**

This paper proposes video-SALMONN S, a streaming audio-visual LLM designed for long-form videos under a fixed memory budget. The core idea is a $\text{TTT}_{\text{MEM}}$ layer that applies Test-Time Training (TTT) to update fast weights, allowing the model to compress historical visual information into parameters instead of storing all past tokens. The model also includes a modality-aware memory reading mechanism to retrieve relevant KV caches conditioned on the query. In addition, the authors introduce ELViM, a benchmark for evaluating episodic learning in long video streams. Experiments show improvements over several baselines.

**Compliance With Llm Reviewing Policy:**

Affirmed.

**Final Justification:**

Considering that most of the concerns have been addressed, I have increased the score from 3 to 4.

I hope the authors can incorporate the rebuttal discussions into the revision.

**Key Questions For Authors:**

The paper tackles the important problem of long-stream video understanding under memory constraints and proposes a TTT-based memory mechanism with promising empirical results. However, the methodological novelty appears somewhat incremental, and the evaluation is missing comparisons with several recent KV cache methods. Some parts of the content need to be improved.

**Limitations:**

Please see weakness and key questions.

**Strengths And Weaknesses:**

Strengths

1. **Introduction of TTT**.  Applying Test-Time Training to long-stream video understanding is an interesting idea. Using fast-weight updates as a form of implicit memory could be a useful alternative to token-level or KV-level compression approaches for streaming models with tight memory constraints.

2. **Benchmark contribution**. The proposed ELViM benchmark focuses on episodic learning in long video streams, which is not well covered by existing benchmarks. Tasks that require recalling procedural information observed much earlier in the video are relevant for evaluating long-horizon multimodal reasoning.

3. **Empirical improvements**. The method shows consistent gains over the selected baselines across several long-video benchmarks and on ELViM, suggesting that the proposed memory mechanism is helpful in practice.

Weaknesses

1. **Limited methodological novelty**.  While the application of TTT to streaming video is interesting, the underlying method appears somewhat incremental. The proposed $\text{TTT}_{\text{MEM}}$ layer mainly extends the standard TTT framework[1] with a long-span prediction objective. In addition, the modality-aware memory reading module seems closely related to AdaReTaKe[2].

2. **Missing KV cache baselines**. The evaluation does not include recent methods that specifically focus on KV cache in methods for streaming video understanding. Most baselines considered here (e.g., Similarity Merging, PEMF) focus on the input token rather than the KV cache. Comparing against approaches such as ReKV[3], StreamKV[4], StreamingTOM[5], and StreamMem[6] would help better position the contribution.

3. **Limitations of the ELViM evaluation**. ELViM mainly uses multiple-choice questions, which may reduce the difficulty of the task. In such settings, models can sometimes rely on recognition or option matching instead of actually recalling procedural information from memory. Including quantitative generative evaluations (e.g., ROUGE-L, CIDEr, or LLM-as-a-judge style metrics) would make the episodic learning claim more convincing.

4. **Clarity issues**. Some parts of the method are hard to follow. In particular, the role of the projection weights ($\theta$) is not entirely clear. The formulation in Equation (8) appears inconsistent with the accompanying textual explanation.

**Reference**

[1] Dalal, Karan, et al. "One-minute video generation with test-time training." Proceedings of the Computer Vision and Pattern Recognition Conference. 2025.

[2] Wang, Xiao, et al. "AdaReTaKe: Adaptive redundancy reduction to perceive longer for video-language understanding." Findings of the Association for Computational Linguistics: ACL 2025. 2025.

[3] Di, Shangzhe, et al. "Streaming video question-answering with in-context video kv-cache retrieval." arXiv preprint arXiv:2503.00540 (2025).

[4] Chen, Yilong, et al. "Streamkv: Streaming video question-answering with segment-based kv cache retrieval and compression." arXiv preprint arXiv:2511.07278 (2025).

[5] Chen, Xueyi, et al. "Streamingtom: Streaming token compression for efficient video understanding." arXiv preprint arXiv:2510.18269 (2025).

[6] Yang, Yanlai, et al. "Streammem: Query-agnostic kv cache memory for streaming video understanding." arXiv preprint arXiv:2508.15717 (2025).

---

> ### Author Rebuttal · Authors · 2026-03-30
>
> We appreciate the reviewer for acknowledging our contribution and providing constructive feedback. We would like to address each individual concerns as follows:
>
> > Weakness 1: While the application of TTT to streaming video is interesting...seems closely related to AdaReTaKe[2].
>
> We agree that our work builds on prior TTT and KV-reading ideas, but the contribution is not a minor variant of either. The paper is, to our knowledge, the __first streaming video LLM to use TTT as the core long-term memory mechanism__, and the proposed system is not just “standard TTT + one extra loss.” It combines:
> - a TTTMEM layer that turns fast weights into an implicit streaming memory
> - a novel long-span prediction objective specifically designed to counter forgetting over long horizons
> - a novel two-stage training scheme that enables scaling context and memory capacity
> - a novel modality-aware prompt-dependent memory reader that operates under strict streaming constraints with constant memory.
>
> > Weakness 2: The evaluation does not include recent methods that specifically focus on KV cache...would help better position the contribution.
> - We have discussed KV cache compression methods in section 2.2, and will include additional papers mentioned in the review in the related work section.
> - We now apply StreamMem, StreamingTOM and AdaReTaKe to the baseline non-streaming video-SALMONN 2+ model, which was built in a setting similar to video-SALMONN S. The results are shown as follows.
> | Model | VMME long | LVBench | VidEvalPro | ELViM |
> |---|---|---|---|---|
> | Similarity Merging | 69.4 | 51.6 | 51.8 | 35.4 |
> | StreamMem | 69.8 | 52.4 | 53.7 | 33.8 |
> | AdaReTaKe | 70.2 | 54.3 | 54.5 | 41.5 |
> | StreamingTOM | 66.7 | 48.9 | 48.1 | 30.5 |
> | video-SALMONN S | 71.3 | 55.4 | 55.8 | 43.9 |
> | video-SALMONN S + Reading | __71.3__ | __55.6__ | __58.9__ | __46.7__ |
>
> __Setup__:
> - For StreamMem, we keep a KV cache size M=16k.
> - For StreamingTOM, we use a group size of 50, a maximum of 960 groups (equivalent to 48k tokens), and de-quantise 320 groups at a time. We also apply the same modality mask to avoid model failure.
> - We were unable to find the official StreamKV implementation or a replication of it. ReKV introduces an external CLIP model for selection.
>
> As shown in the table, KV Cache compression only achieved slightly better performance than the similarity merging baseline, and thanks to the test-time trained memory embedding, video-SALMONN S achieved superior performance compared to all KV-Cache compression methods evaluated above. We will include these results in the revised paper.
>
> > Weakness 3: ELViM mainly uses multiple-choice questions...would make the episodic learning claim more convincing.
> - We select MCQ as the main evaluation format because it bears the advantage of a direct and standard evaluation metric and pipeline. MCQ has been widely adopted as the main format among various video benchmarks.
> - We provide the evaluation based on open-ended questions using Gemini-2.5-Pro with video inputs, as the judge below, which shows the same trend as the MCQs. We will add this table to the revised paper.
> | Model | ELViM MCQ (Acc) | ELViM Open-ended (Acc) |
> |---|---|---|
> | Non-streaming Baseline | 32.5 | 17.8 |
> | Similarity Merging | 35.4 | 22.6 |
> | video-SALMONN S + Reading | 46.7 | 30.2 |
> - We also provided open-ended examples in Appendix C, Figures 10-12 in the submitted paper.
>
> > Weakness 4: Some parts of the method are hard to follow...accompanying textual explanation.
>
> Thank you for the careful reading. We will include the following modifications in the revised paper.
> - Projection weights follow the standard TTT-MLP [1] implementation to establish the multi-view reconstruction loss.
> - We will correct the typo "ArgTop-> ArgTopK" in Eqn. (8).
>
> [1] Y. Sun et al. "Learning to (Learn at Test Time): RNNs with Expressive Hidden States", https://arxiv.org/abs/2407.04620

---

> > ### Author Rebuttal · Reviewer_959h · 2026-04-03
> >
> > Thank you for your response. Most of my concerns are addressed.

---

### Official Review · Reviewer_yaM3 · 2026-03-13

**Soundness:** 3
**Presentation:** 3
**Significance:** 3
**Originality:** 3
**Overall Recommendation:** 4
**Confidence:** 4

**Summary:**

This paper introduces video-SALMONN S, a memory-enhanced streaming audio-visual large language model designed to overcome the memory and information loss challenges in long-video understanding (up to 3+ hours). By incorporating a novel Test-Time Training mechanism, the model transforms video sequences into implicit parameter-based memory via a specialized $TTT_{MEM}$ layer with long-span prediction objectives, maintaining a constant memory budget while preserving long-term context. Supported by a prompt-dependent modality-aware memory reading mechanism and evaluated on the new ELVIM benchmark, video-SALMONN S demonstrates superior performance in learning and recalling procedural knowledge from extended video durations compared to existing non-streaming models.

**Compliance With Llm Reviewing Policy:**

Affirmed.

**Key Questions For Authors:**

* While the model maintains a constant memory footprint, Test-Time Training (TTT) requires constant gradient updates (fast-weights) for every video segment during inference. How does the inference latency and power consumption compare to traditional KV-caching or token compression methods? Is it feasible for real-time deployment on edge devices?
* Since video information is compressed into a fixed-size parameter space ($TTT_{MEM}$ layer), is there a "saturation point"? At what video duration (e.g., 10+ hours) does the model's ability to represent past information begin to degrade significantly?
* Another popular approach for long-context video is Retrieval-Augmented Generation (Video-RAG), which stores frames in an external vector database. What is the core advantage of implicit parameter memory over explicit retrieval in terms of accuracy and interpretability?

**Limitations:**

yes

**Strengths And Weaknesses:**

Strengths:
* The integration of Test-Time Training (TTT) as a memory mechanism is technically sound. The authors address the "forgetting" issue inherent in standard TTT by introducing a long-span prediction objective, which is a principled way to ensure the model retains past information. The paper evaluates the model on a wide range of benchmarks, including Video-MME, LVBench, and MLVU. The consistent 3-7% improvement over both streaming and non-streaming baselines suggests that the performance gains are robust and not overfitted to a specific dataset.
* Introducing the Episodic Learning from Video Memory (ELVIM) benchmark is a significant contribution. It shifts the evaluation focus from mere "perception" (what is happening now?) to "procedural learning" (how do I do what I saw two hours ago?), which is a more demanding and realistic test of AI intelligence.


Weaknesses:
* While the memory is constant, the computational cost of performing gradient updates (fast-weights) for every frame during inference is significantly higher than standard forward-pass inference. The paper could provide a more detailed analysis of the latency trade-offs.
* While the combination is highly original, many individual components are existing technologies. The novelty lies more in the architectural engineering and the training strategy than in a brand-new fundamental learning algorithm.

---

> ### Author Rebuttal · Authors · 2026-03-30
>
> We appreciate the reviewer for the positive feedback and invaluable suggestions. We provide our responses to individual concerns as follows:
>
> > Weakness 1 & Question 1: While the memory is constant, the computational cost...analysis of the latency trade-offs. How does the inference latency...feasible for real-time deployment on edge devices?
> - The paper reports in Appendix D that adding TTTMEM increases FLOPs by only __0.000235 TFLOPs__ on top of a __3.15 TFLOP__ Qwen3-VL 8B forward pass (0.0075% #calculations), i.e., a negligible fraction of total compute.
> - Runtime measurements in Table 3 on a single H800 for 1-FPS, 3600-frame videos show __10.4s__ for similarity merging, __10.4s__ for merging (traditional compression) and PEMF, __10.5s__ for video-SALMONN S (increases forwarding time cost by __0.94%__).
> - Moreover, including TTT will not cause additional first-token latency since TTT encodes the input stream on-the-fly before the prompt is received in streaming.
>
> > Weakness 2: While the combination is highly original...fundamental learning algorithm.
>
> Thank you for acknowledging the novelty of our final model. We want to clarify that, in addition to the novelty in the combination, our modification of the TTT_MEM layer tailored for parameter-based video memory contains three non-trivial structural and implementation innovations, including:
> - A novel long-span prediction objective for TTT learning (Section 3.1).
> - A new two-stage scale-up training scheme dedicated to longer inputs with TTT_MEM as memory (Section 3.2).
> - A novel modality-aware memory reading mechanism as a __dedicated__ KV Cache reading method for TTT_MEM memory to enable larger memory capacity (Section 3.3).
>
> > - Question 2: Since video information is compressed...begin to degrade significantly?
> We provided the experimental analysis of memory decay in Figures 7 and 8 in the submitted paper.
> - In Figure 7, we showed that TTT_MEM reaches the __plateaus around 8000 frames__, corresponding to __~700k__ input tokens under 1 FPS and 360p. TTT_MEM maintains the performance level at __8192 frames__, whereas merging baselines decayed sharply.
> - In Figure 8, we showed the temporal distance between the question and the answer, up to a span of over 2 hours (__500k tokens__). TTT_MEM __sustains high accuracy__, indicating the retention of information.
> - The exact duration plateau onset depends on the frame rate and resolution, which can be computed directly from our provided token budgets.
> - We further provide performance on ~10-hour videos by concatenating each video in LVBench with two other videos from LVBench, where the answer is in the first video. (Note that both 3-hour and 10-hour settings are generalization to long sequences since during training the maximum length is 2048 frames and ~30 mins)
> | Model | LVBench | LVBench ~10-hour (32k frames, ~2.5M input tokens) |
> |---|---|---|
> | Similarity Merging | 51.6 | 42.3 |
> | video-SALMONN S | __55.4__ | __49.8__ |
> Forgetting gets faster when the duration increases to 10+ hours of ultra-long videos, but it is __significantly mitigated__ compared to the merging baseline. We will make the above analysis clearer in the revised paper.
>
> > - Question 3: Another popular approach for long-context video is Video-RAG...accuracy and interpretability?
> Implicit parameter memory has the following advantages:
> 1. Parameter memory performs __continuous online integration__ under a __fixed memory budget__, rather than storing explicit past frames with memory requirements __growing linearly__ and unbounded.
> 2. The retrieval process usually results in __higher latency__ since the retrieved video needs to be loaded into the GPU memory and then processed by the video encoder and LLM. This may cause a __significant increase in first token latency__, depending on the data transmission bandwidth and how many frames are retrieved.
> 3. Explicit retrieval sees only what was stored and later retrieved; any missed evidence is __effectively unavailable at answer time__. TTT_MEM, instead, integrates each incoming segment into a continuously updated implicit memory, so that history influences the model even when raw past tokens are no longer retained. Thus, the advantage is not an exact replay of the whole past, but online consolidation of the full stream without a separate retrieval failure mode

---

> > ### Author Rebuttal · Reviewer_yaM3 · 2026-04-03
> >
> > Thank you to the authors for the rebuttal. The responses address my main concerns. I will maintain my rating.

---

### Official Review · Reviewer_YuwN · 2026-03-17

**Soundness:** 2
**Presentation:** 4
**Significance:** 3
**Originality:** 3
**Overall Recommendation:** 3
**Confidence:** 4

**Summary:**

This work presents video-SALMONN S, a framework designed to facilitate the real-time understanding of long-duration video streams exceeding three hours. To address the pervasive issues of information loss and memory constraints in existing models, the authors introduce TTT as a streaming memory mechanism, which is a first in the field of video understanding. The technical contribution is centered on a two-stage training scheme optimized for TTT parameters and a modality-aware memory reader that selectively retrieves relevant context based on user prompts.

**Compliance With Llm Reviewing Policy:**

Affirmed.

**Key Questions For Authors:**

1. Given the additional FLOPs and power consumption required for gradient calculations at every frame or batch during inference, can you provide a more detailed analysis of the model's sustainability for mobile agents or real-time embedded systems?

2. Since audio tokens bypass the TTT memory and are passed directly, how does the model maintain temporal alignment with the video memory in scenarios where audio cues are the primary drivers of context?

3. Have you conducted stability tests or convergence proofs for the TTT_MEM layer parameters during ultra-long streams? I would appreciate further evidence that the parameter values do not diverge over extended horizons.

**Limitations:**

yes

**Strengths And Weaknesses:**

The paper offers an elegant solution to the linear growth of KV caches by re-framing memory as a dynamic parameter update process, significantly enhancing memory efficiency. Beyond standard QA tasks, the introduction of an agent-centric evaluation benchmark is a forward-looking contribution. Furthermore, the model demonstrates a clear competitive edge, consistently outperforming current SOTA baselines under identical memory budgets.

A primary concern lies in the computational overhead since TTT necessitates backpropagation during the inference phase, it inherently demands substantially more FLOPs than a standard Transformer forward pass. While the authors emphasize memory efficiency, the manuscript lacks a rigorous, hardware-level analysis of real-time inference latency. Additionally, there is a non-trivial risk of catastrophic forgetting or overfitting to recent frames as sequences extend into tens of hours. The paper would be strengthened by a more granular theoretical or experimental analysis of memory decay to address potential information loss in the early parts of ultra-long streams.

---

> ### Author Rebuttal · Authors · 2026-03-30
>
> We appreciate the reviewer for acknowledging our contribution and providing constructive feedback. We would like to address each individual concern as follows:
>
> > Weakness 1: A primary concern lies in the computational overhead...hardware-level analysis of real-time inference latency.
> - We would like to point out that the TTT inner-loop back-propagation is implemented as a forward pass by computing the __closed-form gradients__ and performing __one iteration of gradient descent__. Therefore, this layer consumes negligible computation overhead in theory. We provided the __full implementation details__ in the supplementary material.
> - Moreover, we have conducted hardware-level analysis:
>   - Appendix D shows that TTTMEM adds only __0.000235__ TFLOPs on top of a 3.15 TFLOP Qwen3-VL 8B forward (0.0075%).
>   - In Table 3, we measured the run-time statistics. The additional inference time cost using TTT is negligible (__0.1s out of 10.4s__) compared to the baseline. This number is around 200k input tokens, so per-token latency is in the microsecond range.
>   - Moreover, including TTT will not cause additional first-token latency, since TTT encodes the input stream on-the-fly before the prompt is received.
>
> > Weakness 2: Additionally, there is a non-trivial risk of catastrophic forgetting...in the early parts of ultra-long streams.
> - We provided the experimental analysis of memory decay in Figures 7 and 8 in the submitted paper.
> - In Figure 7, we showed that TTT_MEM reaches the __plateau at around 8000 frames__, corresponding to ~700k input tokens under 1 FPS and 360p. TTT_MEM maintains the performance level at __8192 frames__, whereas merging baselines decayed sharply.
> - In Figure 8, we showed the temporal distance between the question and the answer in the video, up to a span of 2 hours (500k tokens). TTT_MEM __sustains high accuracy__, indicating the retention of information.
> - These results directly support the claim that TTT_MEM largely mitigates, rather than exacerbates, long-horizon forgetting compared to the baseline methods.
> - From an optimisation perspective, we only conduct one-iteration gradient descent on each input token, hence overfitting is unlikely to happen.
> - We further provide performance on ~10-hour videos by concatenating each video in LVBench with two other videos from LVBench, where the answer is in the first video. (Note that both LVBench standard and 10-hour settings are generalization to long sequences since the maximum length seen in training is 2048 frames and ~30 mins)
> | Model | LVBench | LVBench ~10-hour (32k frames, ~2.5M input tokens) |
> |---|---|---|
> | Similarity Merging | 51.6 | 42.3 |
> | video-SALMONN S | __55.4__ | __49.8__ |
> Forgetting gets faster when the duration increases to 10+ hours of ultra-long videos, but it is __significantly mitigated__ compared to the merging baseline. We will make the above analysis clearer in the revised paper.
>
> > Question 1: Given the additional FLOPs...real-time embedded systems?
> - The hardware-level evidence in the paper suggests the main bottleneck is video encoding, while TTTMEM contributes only ~0.1s overhead over a ~10s pipeline for a 1-FPS, 3600-frame sample. TTTMEM only takes 0.0075% of calculations and 0.94% of the time cost compared to the Qwen3-VL 8B inference cost, which is already practical for both real-time server-side and mobile-side agents.
>
> > Question 2: How does the model maintain temporal alignment...are the primary drivers of context?
> - We keep a record of the position of each audio token, and insert the downsampled audio tokens back into the input stream, keeping the same relative position to the visual and text tokens. We will make this clearer in the revised paper.
>
> > Question 3: Have you conducted stability tests...ultra-long streams? ...extended horizons.
> - Below, we report $||\theta_{t+1}-\theta_{t}||_2$ to show that it does not diverge on the long horizon up to __10M__ tokens by concatenating 3 LVBench videos together, where $\theta_t$ is the TTT fast weight.
> | Token Position | 100k | 500k | 1M | 2M | 5M | 10M |
> |---|---|---|---|---|---|---|
> | $\|\|\theta_{t+1}-\theta_{t}\|\|_2$ | 0.181 | 0.053 | 0.288 | 0.208 | 0.031 | 0.105 |
> | $\|\|\theta_{t}\|\|_2$ | 28.85 | 27.55 | 27.33 | 29.94 | 29.51 | 29.22 |
> We observe slightly larger gradient norms in the first 10k tokens, and then reach a steady value.

---

> > ### Author Rebuttal · Reviewer_YuwN · 2026-04-05
> >
> > Thank you for the rebuttal, I will keep my original score.

---

### Decision · Program_Chairs · 2026-04-30

**Decision:**

Accept (regular)

**Comment:**

After all reviewers have acknowledged the rebuttal, this paper received mixed ratings, including one weak reject and three weak accept. Though the final rating of Reviewer eRWw is weak reject, he/she does not present a detailed concern after rebuttal. Its merits, including the enhancement of memory efficiency, integration of Test-Time Training, and the proposal of a new benchmark, are well recognized by the reviewers. As for the weakness, the rebuttal addresses most of the concerns.

I think the current manuscript is ready for publication. Please incorporate the response into the revised manuscript.